# Revealing the Complex Interplay of Biostimulant Applications

**DOI:** 10.3390/plants13162188

**Published:** 2024-08-08

**Authors:** Ye Yuan, Nicholas Dickinson

**Affiliations:** 1Faculty of Agriculture and Life Sciences, Lincoln University, Christchurch 7647, New Zealand; nicholas.dickinson@lincoln.ac.nz; 2High Country Salmon, Glenbrook, Twizel 7999, New Zealand

**Keywords:** horticulture, humic acids, protein hydrolysates, seaweed extracts, *Antirrhinum*

## Abstract

Some biostimulant products provide proven benefits to plant production, potentially offering more environmentally friendly, sustainable, and natural inputs into production systems. However, the transference and predictability of known benefits between different growth environments, application protocols, and management systems are fraught with difficulty. In this study, we carried out carefully controlled glasshouse and in vitro assays with applications of humic acids, protein hydrolysates, and seaweed extract to compare the variability of biostimulant effects and dosage-dependent variations across diverse conditions, encompassing a sufficient range to comprehensively assess their full spectrum of impacts. The results demonstrated a clear trend of dosage-dependent effects with each biostimulant exhibiting a significant growth-promoting effect within a critical concentration range, but detrimental effects when the concentration fell outside this range. While substantial growth-promoting effects were observed under glasshouse conditions, biostimulant applications tended to be more sensitive and generally led to negative impacts in sterilised conditions. The combined use of biostimulants mostly resulted in detrimental and toxicological responses with only two combined treatments showing marginal synergistic effects. The findings demonstrated a complex interplay between biostimulants and the growth conditions of plants. Lack of knowledge of the indirect effects of different growth media may result in negative impacts of biostimulant applications and combinations of products outside narrow critical concentration ranges.

## 1. Introduction

In recent years, biostimulants have emerged as a compelling novel approach for sustainable crop production, driven by the demand for eco-friendly agricultural inputs and the opportunity for optimum productivity [1]. Surprisingly, they have made few inroads into horticultural industries where there is high public visibility and significant commercial and public demand for organic products, including in the production of garden plants, house plants, and cut flowers. Unlike conventional inputs such as inorganic fertilisers, biostimulants have more influence on plant metabolism rather than, in this case, simply providing a source of nutrients. In the recent European Union (EU) Fertilising Products Regulation, a biostimulant is defined as “a product stimulating plant nutrition processes independently of the product’s nutrient content, with the aim of improving one or more of the following characteristics of the plant: nutrient use efficiency, tolerance to abiotic stress, crop quality traits or availability of confined nutrients in the soil and rhizosphere” [2,3]. There is a wide range of potential biostimulants, although humic acids, protein hydrolysates, and seaweed extracts account for the most studied non-microbial biostimulants and are the most commercialised in the market [4].

The beneficial effects of biostimulants have been widely reported and documented, but application rates and methodologies vary, frequently making the results difficult to compare and reproduce, and thus unpredictable and unreliable [5,6]. Many factors, such as the source of the product, environmental conditions, mineral nutrient content of the growth medium, and the species of plants have been found to affect the performance of biostimulants creating difficulties for users [7,8]. Although most research focuses on edible crops, biostimulants have also shown efficacy in enhancing plant biomass, accelerating growth and flowering, improving plant quality, and shortening the cultivation cycle for many ornamental species [9]. Among the diverse varieties of ornamental plants, *Antirrhinum majus* L. is recognised as one of the most promising model plant species, which has been widely used for genetic and molecular studies [10]. However, there are only a few published research articles investigating the effects of biostimulants on *Antirrhinum majus* L. [11,12]. The limitation on the number of research articles, combined with variations in biostimulant types and environmental conditions, makes the results of current studies incomparable. There is clearly a need to investigate the effects of a broader range of biostimulants on *Antirrhinum majus* L. 

A systematic study is also required to investigate the interactions between different categories of biostimulants, despite several reports of additional benefits through using multiple types of biostimulants. Commercial products consisting of more than one type of biostimulant are readily available in the market, even though the proven effects of combined biostimulants seem to be elusive [13,14,15,16]. In fact, combinations of two or more biostimulants have been found to be additive, synergistic, or antagonistic interactions, and different to the effects that obtained from individual products [17,18]. On the other hand, the effects of biostimulants are not always positive, with some studies reporting biostimulants have no effects or minimal effects on plant growth [19,20]. This could be due to the variations in the content of biostimulants, plant species, application rate, and environmental conditions. It has been recognised that excessive application of biostimulants may cause adverse effects on plants and the environment [21,22]. Nevertheless, it is still unknown how the dosage effects interact with growth conditions and how plants respond to different types of biostimulants.

The aim of the present study is to evaluate the effects and interactions of the three main categories of biostimulants under carefully controlled glasshouse and in vitro conditions using *Antirrhinum majus* as a model plant with a specific focus on the overall dosage effects.

## 2. Results

### 2.1. Dosage Effects of Greenhouse Conditions

Within a defined range of application rates, the tested biostimulants all showed significant growth-promoting effects, but there were detrimental effects outside this critical range (Figure 1). Visible morphological changes were observed for all the three tested biostimulants under certain concentration ranges (Figure 2a), and dose-dependent effects were visibly most pronounced in seaweed-extract-treated plants (Figure 2b).

Analysis of the nine growth and physiological parameters using PCA showed the first two principal components explained 75.7% of the dataset (Figure 3a). Relative water content (RWC%) was separated by PC 2. The biostimulants had a general positive influence on RWC%, particularly for those within the critical ranges; only SW1 and HA 12 were lower than the control treatment (Figure 3b). Higher PC1 results were evident for SW 0.25, HA 1.5, and PH 0.24, compared to the other treatments (Figure 3b). The optimal application rates for the three biostimulants were 0.25 g L^−1^ for seaweed extract (SW 0.25), 1.5 g L^−1^ for humic acids (HA 1.5), and 0.24 mL L^−1^ for protein hydrolysates (PH 0.24).

### 2.2. In Vitro Experiment

The application of a single biostimulant under in vitro conditions led to detrimental impacts on the growth of *Antirrhinum* seedlings, and the dosage requirement for the biostimulants was substantially diminished compared to the optimal rate found in the glasshouse experiment especially for humic acids (Figure 4).

The results of the principal component analysis also demonstrated the same trend. The growth and development scores across various parts of the plant generally aligned with the PC1 values (Figure 5a). Biostimulant treatments, specifically those at high and low dose levels, were found to have more pronounced detrimental effects on plant growth, as indicated by lower PC1 values, compared to treatments within the medium dose range (Figure 5b).

In terms of the combined effects, only two combinations showed marginal positive effects out of the 54 combined biostimulant treatments (72 biostimulant treatments in total; Figure 6). Visible differences in morphological appearance were observed when humic acids and seaweed extract were combined (0.001% HA + 0.001% SW), and humic acids and protein hydrolysates (0.0001% HA + 0.005% PH). These effects were evident in better shoot and root growth (Figure 7).

Treatments applied with biostimulants at the highest concentration exhibited significant negative impacts on the growth and survival rates of plants, particularly in those treatments where humic acid was present (Figure 4 and Figure 7). However, abnormal plant morphology changes were more frequently observed at extreme low concentrations. In combinations that led to a high mortality rate, deformation was evident in the surviving plants that induced unusual shoot growth, changes in the shape of the leaves, and undeveloped roots (Figure 8).

## 3. Discussion

Dosage effects were extensively observed for all tested biostimulants regardless of growth conditions or whether they were added alone or in combinations. This effect has been previously recorded and described as phytohormone-like activity in the past [23,24].

However, the dosage effects varied significantly under different conditions.

Under greenhouse conditions, most treatments resulted in positive influences on plant growth, with negative impacts observed in a few treatments with a high application rate. This aligns with previous studies that demonstrated that excessive use of biostimulants could cause adverse effects [21,22]. When comparing the dosage effects among different biostimulants, seaweed extract demonstrated a more pronounced dosage-dependent effect under glasshouse conditions followed by humic acids and protein hydrolysates (Figure 3), suggesting that the nature and magnitude of dosage effects are highly dependent on the specific biostimulant used, which may be associated with variations in the nutrient content and biochemical composition of the tested biostimulants [25,26,27,28,29,30,31,32]. Further research is required to determine the specific roles of each of the bioactive components in the overall performance of biostimulants and to investigate potential interactions between different components.

In contrast to glasshouse experiments, the results of in vitro experiments showed a completely opposite trend, with all treatments found to have no effects or negative effects on plant growth. However, the dosage effects were still evident with significant negative growth impacts exhibited for the treatments applying biostimulants at the lower and higher concentration. The biostimulant application rates which showed positive effects under greenhouse conditions apparently overdosed and resulted in detrimental effects under in vitro conditions (Figure 4). Compared to the glasshouse study, the overall dosage effect appeared to be more sensitive to biostimulant addition under the in vitro conditions. The varied effects are likely the consequence of a complex blend of factors such as environmental conditions, age of plant, nutrient content, management system, pH, and other characteristics of the growth medium. However, assessing the individual impact of each factor is challenging, particularly when considering the potential interactions among these factors. Although the present study does not specifically investigate the effects of the influencing factors, the differences in the dosage effects between the two experiments are probably a result of indirect effects of biostimulants interacting with the surrounding enviroment, such as microbial comminities and the property of the growth medium, since the in vitro experiment is designed to test the direct effect of the biostimulant by diminishing these factors.

The dosage effects observed under in vitro conditions are also influenced by the type of biostimulant used. Humic acids exhibited severe negative effects at higher dose rates, which is not surprising considering a number of research articles highlight the detrimental effects of overdosing humic acids [33,34]. However, it appears that the negative effects also occur at much lower doses than what was observed in the greenhouse experiment. The negative impacts observed with excessive application were proposed to be due to humic acid adhering to the pores of plant roots’ cell walls, resulting in obstruction and reduced function [35]. Thus, the finer and more homogeneous nature of Phyto agar, as compared to the potting mix, may be the reason for the observed effects, as humic acids may more easily inhibit root activity and nutrient mobility in this medium. It is noteworthy that not only does the excessive application of biostimulants result in negative impacts, but also applying biostimulants at an extremely low rate causes negative effects on plant growth, particularly for seaweed extracts. Similar effects were also observed in combined biostimulant treatments under a low concentration rate. To the best of our knowledge, the present study is the first to record a negative effect of a biostimulant at low concentrations, as most current studies focus on only two or three levels of biostimulant application rates close to the optimal rate, without encompassing a wide range of dosages to cover the full effects of biostimulants. Although the precise mechanisms behind the results are uncertain, they are likely to be involved with the complex mechanisms of bioactive components contained in the biostimulants [36,37,38]. The findings highlight the risks of biostimulant applications particularly for combined use, and extra caution is required when applying biostimulants in practice.

Although the biostimulant did not show positive effects at the concentration tested under the in vitro experiment, this does not mean that its effects are necessarily negative. As a number of studies have recorded the positive effects under environmentally controlled in vitro conditions, the negative result obtained in our experiment is probably attributed to the fact that the concentration level of the substance under investigation did not fall within the narrow crucial range that is necessary for a positive outcome [39]. Nevertheless, there is no doubt that the optimal dosage found under glasshouse conditions is not suitable to be directly applied under in vitro conditions. Therefore, it is necessary to consider the growth or environmental conditions when applying biostimulants to plants. A better understanding of the interactions between biostimulants, plants, and the environment will be particularly valuable for developing more precise guidelines for biostimulant use in the future.

## 4. Materials and Methods

### 4.1. Plant Materials and Constitution of Biostimulants

*Antirrhinum majus* L. was used as the test plant in all the experiments; F1 Hybrid plants (Cultivar: Snapshot White) were germinated from seeds sourced from the Egmont Seed Company Ltd. (Christchurch, New Zealand). Commercial biostimulants were obtained online or from local retailers: (i) humic acids (70% sodium humate), extracted from leonardite (Czy Delgado, Shijiazhuang, China); (ii) protein hydrolysates, sourced from marine animals (United Fisheries Ltd., Christchurch, New Zealand); (iii) seaweed extract (water-soluble powder) extracted from *Ascophyllum nodosum* (Fertile Field Ltd., New Plymouth, New Zealand). Analyses of trace elements of the three products (Table 1) and the amino acid profiles of the protein hydrolysates (Table 2) were carried out using standard protocols by Lincoln University Analytical Services.

### 4.2. Glasshouse Experiment

Plants were grown under natural light conditions in a glasshouse (temperature range, 15.0 to 27.4 °C, mean 18.5 °C). A commercial potting mix designed for ornamental plants (Tully’s, Lyndale Custom Mix, Auckland, New Zealand) was used as the growth medium (Table 3). The potting mix consists of a slow-release fertiliser to provide the plant with adequate nutrients throughout its lifetime.

Five different application rates were applied for each of the three tested biostimulants, respectively. These rates were determined based on the manufacturer’s recommended dosage, adjusted to correspond to grams or millilitres per litre of potting mix (Table 4). Biostimulant treatments were applied to established *Antirrhinum* plants (5-week-old plants) by transplanting them into 750 mL pots filled with potting media mixed with the biostimulant at varying rates according to the experimental design with six replicates per treatment. The tested plant materials were selected twice prior to biostimulant application to minimise the variance. To obtain the uniform plants, adequate *Antirrhinum* seeds were first sown in vermiculite; two weeks after germination, uniform seedlings were selected and transferred to six-cell seedling pots (60 mL) filled with Tully’s potting mix for a further two weeks. After the acclimatation, uniform plants were then selected again and subjected to the test.

After setting up all the treatments, the pots were then arranged on a glasshouse table in a randomised block design for 60 days. Fifteen plants from different biostimulant treatments (three biostimulants × five application rates) were randomly placed in a plastic pot tray as a single experimental unit, with six units in total. To minimise the influence of any positional effects, the pots within the unit were re-randomised every week, and the relative positions of the trays were also changed. Throughout the experimental period, all plants were gently watered daily as required, with no water overflow from the pots. Photographs and measurements of plant morphology (number of lateral shoots, leaf number, leaf area, shoot length) were carefully recorded throughout the experimental period. At the end of the experiment, all plants were harvested and the potting medium was gently removed and washed from the roots. The plants were then divided into shoots, leaves, roots, and flowers to calculate the fresh weight, root and shoot length, number of leaves, branches, and flowers. Leaf area was also determined using a Li-Cor 3000 Leaf Area Meter (Li-Cor, Lincoln, NE, USA), and the fresh leaves were then immersed in distilled water for 24 h for further calculation of relative water content RWC % following the method described by Akladious and Mohamed [40]. The dry weights of different parts of the plants were measured by oven-drying plant materials at 65 °C for 48 h.

### 4.3. In Vitro Experiment

A second experiment was carried out in an environmentally controlled growth chamber, under long day conditions (16 h light/8 h dark, 22 °C, 300 μmol m^−2^ S^−1^ of light intensity) using *Antirrhinum* seeds from the same batch as those used in the glasshouse experiment. The combination effect of the three biostimulants was also evaluated in the test. Assays were carried out with seventy-three different combinations and six concentrations of the three biostimulants (including controls). The experimental work was carried out separately for two groups: low concentration (0.0001–0.005%) and high concentration (0.01–0.1%) rates, with each lasting for 35 days (Figure 9).

The experimental procedure followed Garnica-Vergara et al. [41] (Figure 10). *Antirrhinum* seeds were surface-sterilised using 95% ethanol for 5 min, followed by immersion in 20% (*v*/*v*) bleach for 7 min, then washed five times in distilled water. The sterilised seeds were placed in plastic petri dishes containing a 0.2 strength Murashige and Skoog Basal Salts Medium with vitamins (Phyto Technology Laboratories, Lenexa, KS, USA) in Phyto agar (5 g L^−1^, Duchefa Biochemie, Haarlem, The Netherlands). After five days, the germinated seedlings were transplanted to square petri dishes (120 mm × 120 mm × 17 mm) containing the same medium mixing with various biostimulants in different concentrations, within a laminar flow cabinet. Each square petri dish contained four seedlings, with three replicate dishes per treatment. They were then randomly positioned in a growth chamber with a 16 h light/8 h dark cycle, maintained at 22 °C, with a light intensity of 300 μmol m^−2^ s^−1^ during the light period. To encourage root growth on top of the agar, the plates were inclined at 65° using plastic petri dish holders [42]. A fast assessment method was developed to quantify the growth of the shoots and roots of the 876 seedlings (Table 5). Mean growth scores of each subcategory were calculated at the end of the experiment.

### 4.4. Statistics Analysis

Data were analysed using JMP (SAS institute, Cary, NC, USA). One-way analysis of variance was used to test differences between biostimulant effects under different application rates, separating means using post-hoc multiple comparison tests (Tukey’s HSD test). One-way ANOVA was also used to analyse the data of the in vitro experiment, and Dunnett’s test was used to compare the means of biostimulant treatments with the untreated controls. Multivariate analysis (PCA) was used to investigate relationships between the various growth parameters. A heatmap chart was then generated using Microsoft Excel, Version 16.87, to visually represent the effects.

## 5. Conclusions

The study provided insights towards explaining the lack of consistency in the responses of plants to the three main groups of biostimulants under diverse environmental conditions. Dosage effects were observed regardless of growth conditions. While positive effects were generally found within the optimal range under greenhouse conditions, biostimulant application generally resulted in negative growth impacts under sterile conditions. Plants are more sensitive to the application of biostimulants with less of a dosage requirement at the seedling stage of growth and when grown in a less complex growth medium with less potentially confounding factors. Accurately predicting the impacts of biostimulants requires a better knowledge of the indirect effects of different growth media and environmental conditions. The present findings indicate the complexity of combined biostimulant uses and associated environmental influences. Extra caution is required when applying biostimulants in practice, and it is recommended to conduct a small-scale trial before broader application.

## Figures and Tables

**Figure 1 plants-13-02188-f001:**
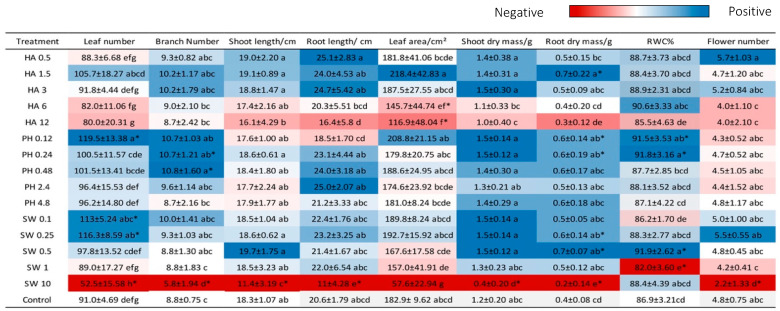
Growth of plants in the glasshouse experiment. Heatmap indicates positive and negative effects of the biostimulants. Different letters indicate significant differences (*p* < 0.05) within each treatment (Abbreviation code defined in Section 4). Asterisks indicate significant differences between control and biostimulant treatments *p* ≤ 0.05.

**Figure 2 plants-13-02188-f002:**
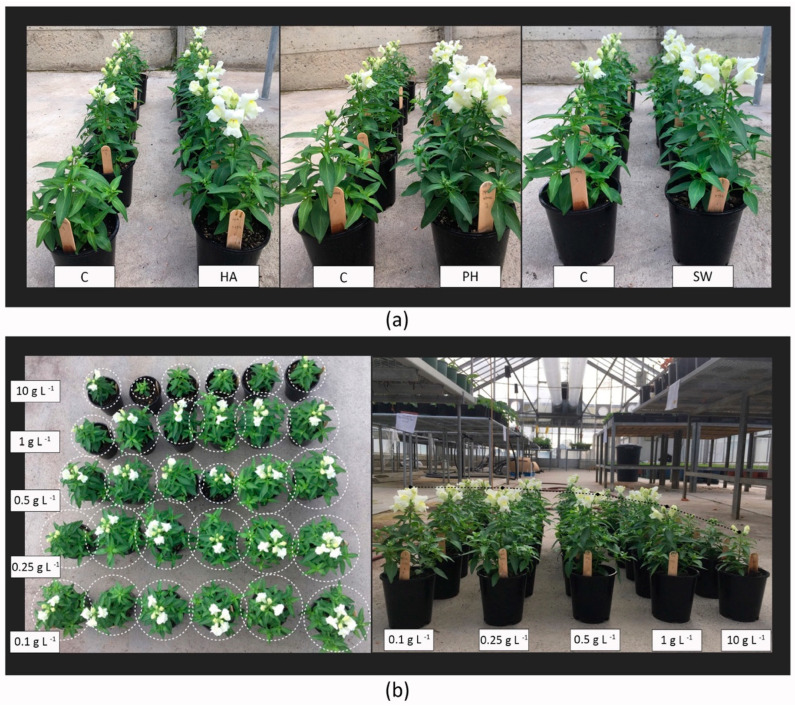
Visible morphological changes showing (**a**) Comparison of morphological appearance between control and biostimulant-treated plants; (**b**) Dosage effects of seaweed extract on the morphological appearance of *Antirrhinum majus* (C: control; HA: humic acid; PH: protein hydrolysates; SW: seaweed extract).

**Figure 3 plants-13-02188-f003:**
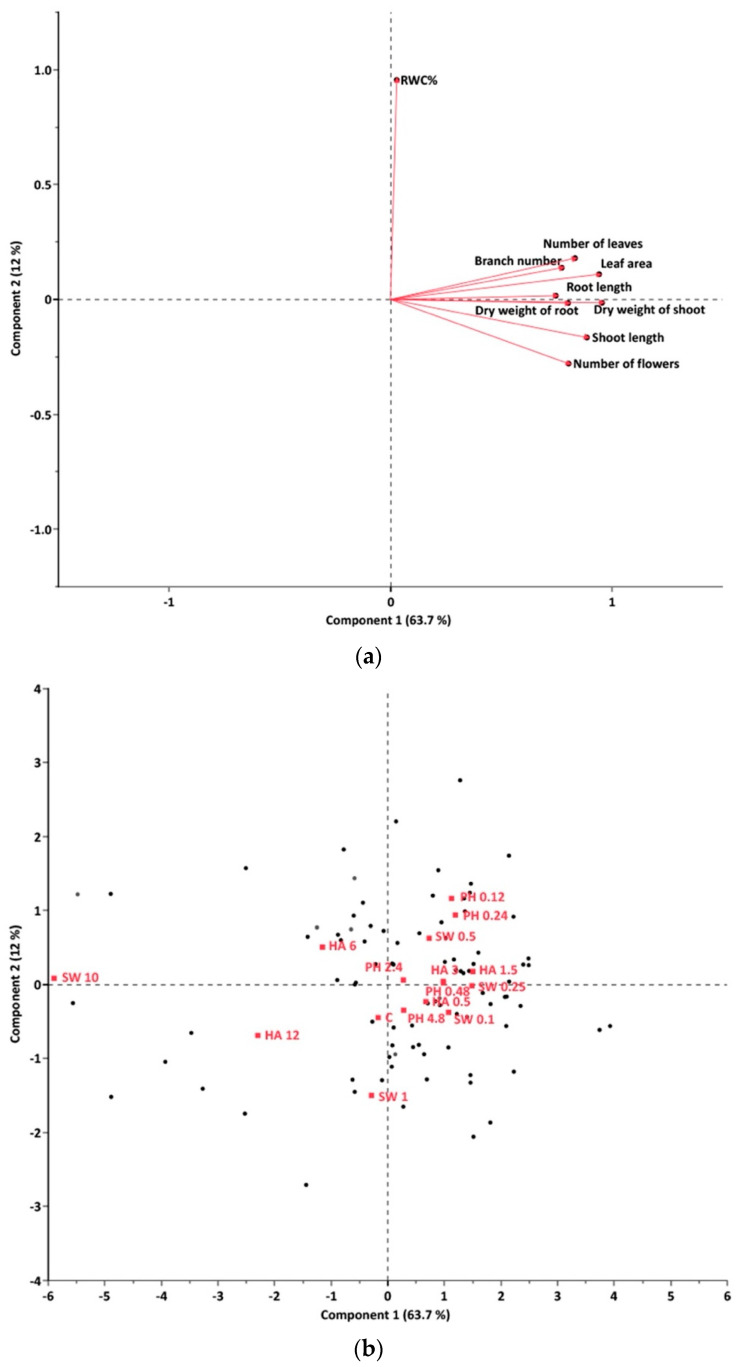
PCA separation of growth and physiological responses to biostimulants in the glasshouse experiment; (**a**) Loading plot; (**b**) Score plot (Abbreviation codes defined in Section 4).

**Figure 4 plants-13-02188-f004:**
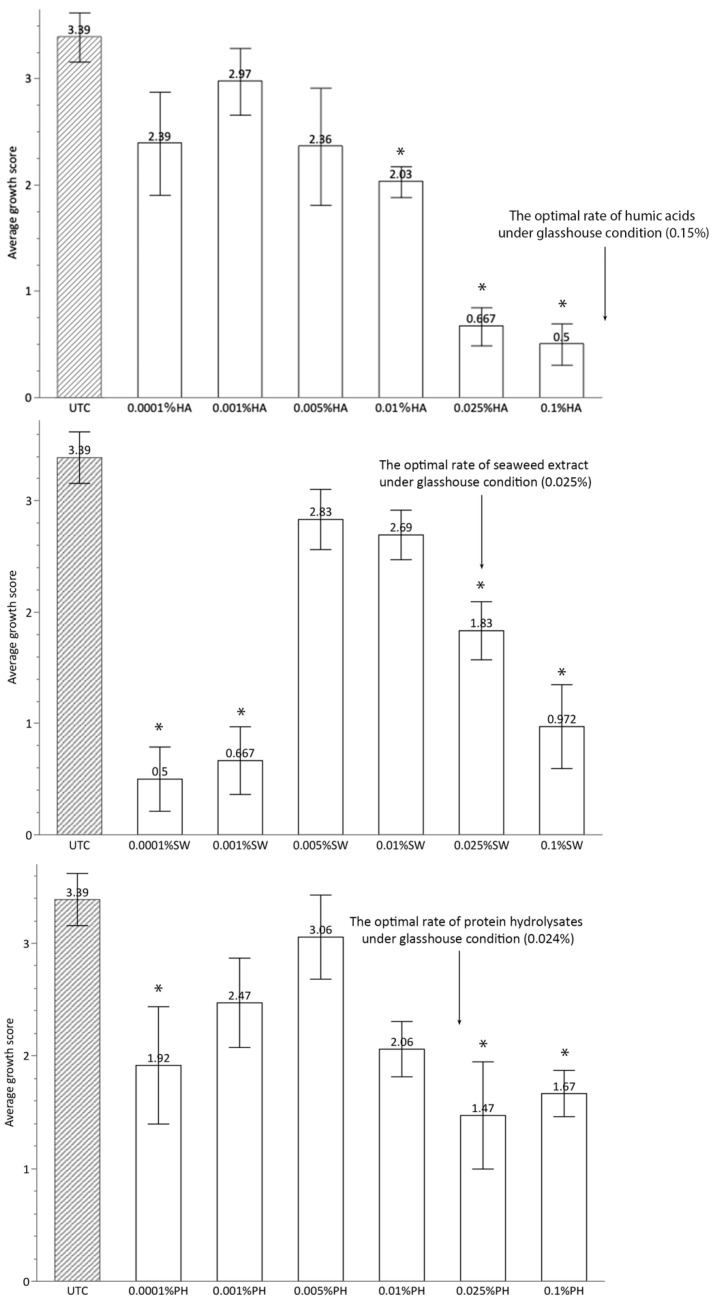
Growth responses of three biostimulants under in vitro conditions (The average growth score is the mean of scores from various aspects of plant growth following a divided scoring criteria; the bar displays the standard errors; Asterisks indicate significant differences between untreated control and biostimulant treatments *p* ≤ 0.05).

**Figure 5 plants-13-02188-f005:**
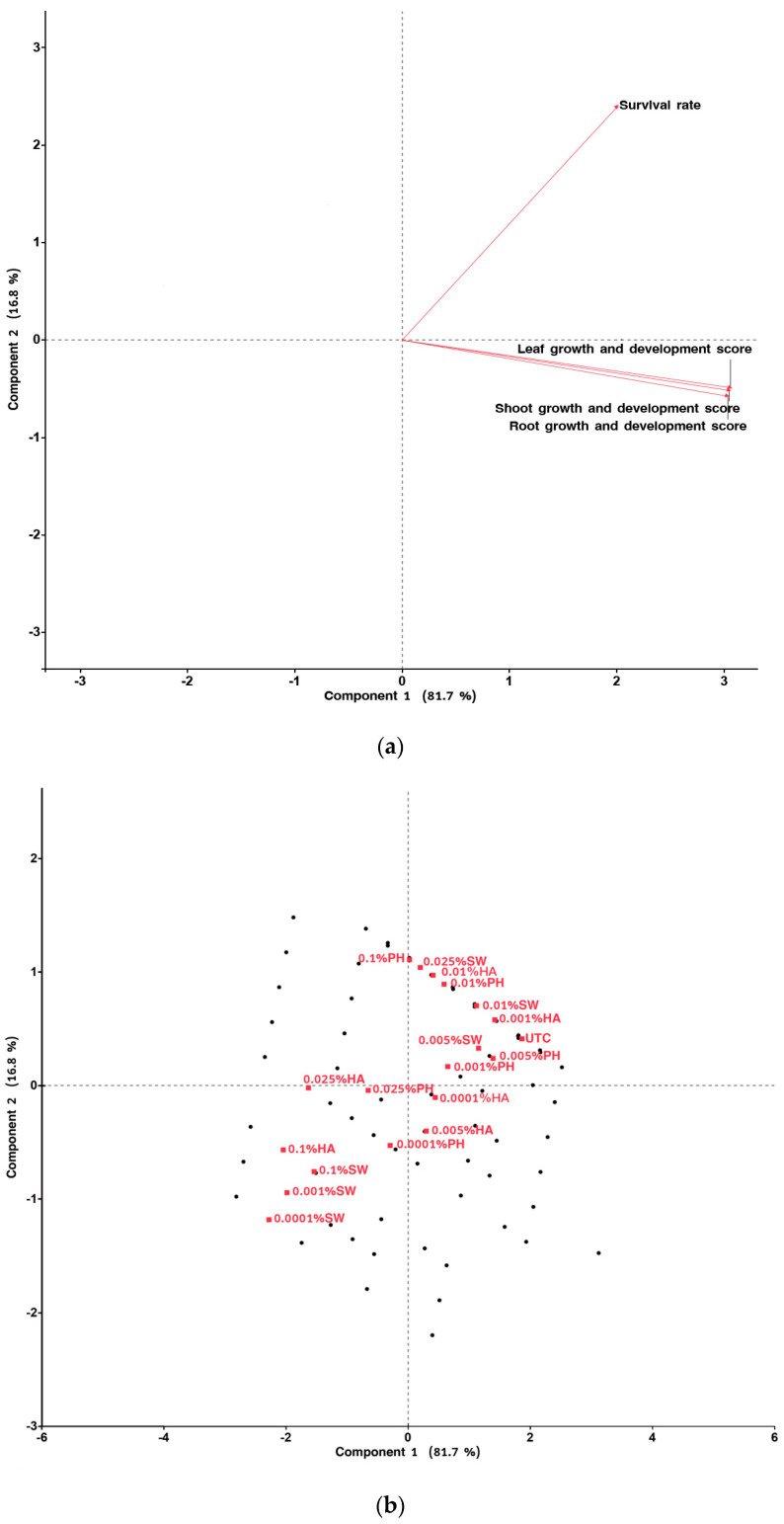
PCA separation of growth and survival responses to biostimulants under in vitro conditions; (**a**) Loading plot; (**b**) Score plot (Abbreviation codes defined in Section 4).

**Figure 6 plants-13-02188-f006:**
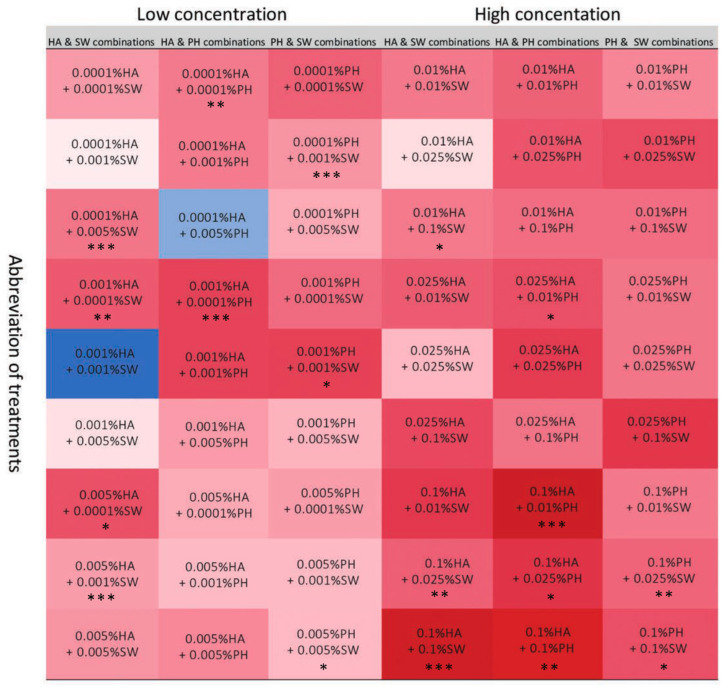
Heatmap showing the average growth score of seedlings affected by the application of combined biostimulants under in vitro conditions (Abbreviation defined in Section 4). Colour indicated by average growth score, calculated by the mean scores from various aspects of plant growth following a divided scoring criteria; Asterisks indicate significant differences between untreated control and biostimulant treatments (* *p* < 0.05; ** *p* < 0.01; *** *p* < 0.001).

**Figure 7 plants-13-02188-f007:**
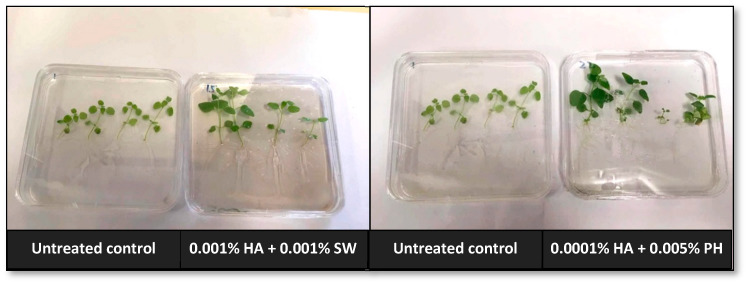
Comparison of morphological appearance in *Antirrhinum* seedlings treated with biostimulants.

**Figure 8 plants-13-02188-f008:**
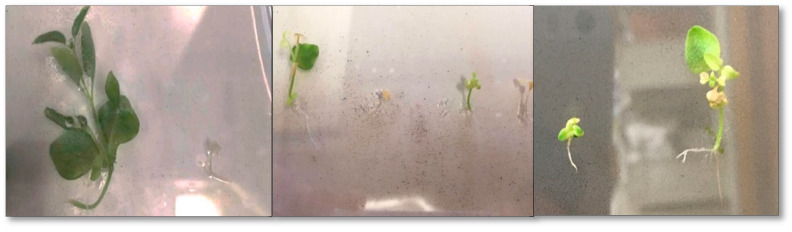
Abnormal growth interrupted by biostimulant applications at low concentration.

**Figure 9 plants-13-02188-f009:**
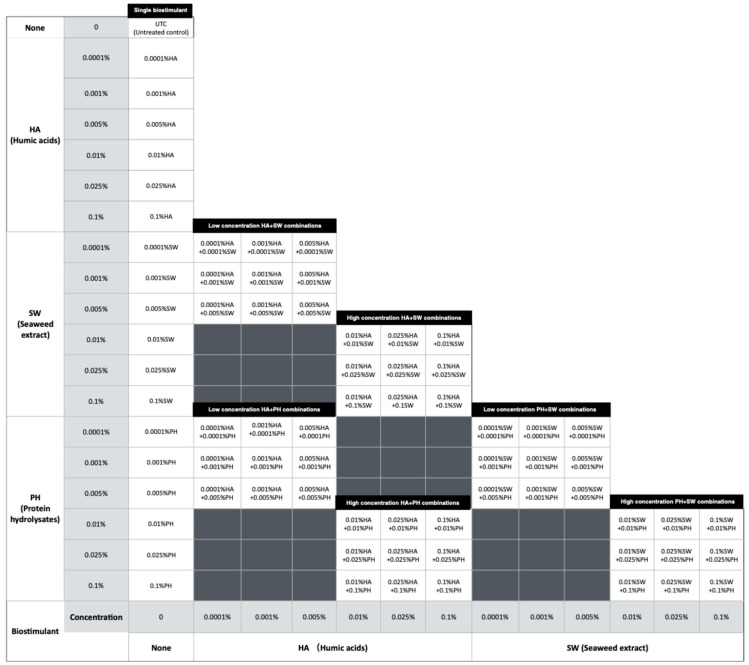
The combinations and abbreviation codes of treatments that provided combinations of different concentrations of each biostimulant under in vitro conditions.

**Figure 10 plants-13-02188-f010:**
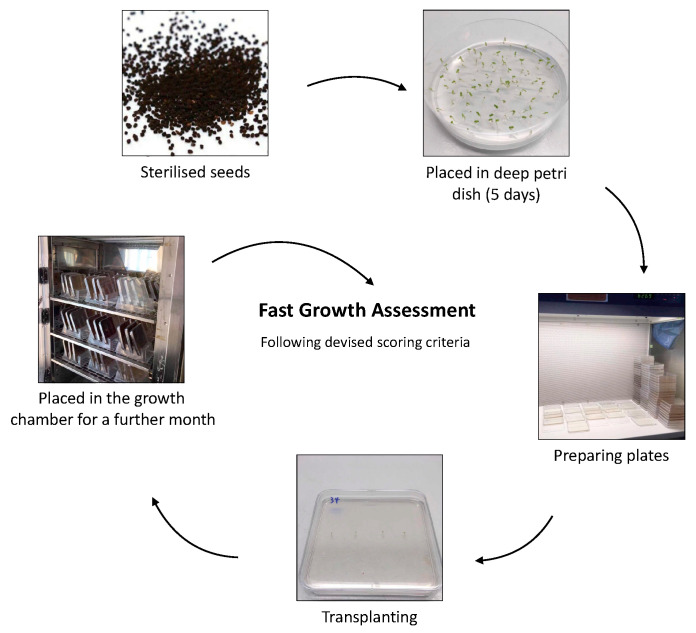
Illustration of the experimental procedure of the in vitro experiment.

**Table 1 plants-13-02188-t001:** Chemical composition of the three biostimulant products.

Component	Humic Acids (HA)	Protein Hydrolysates (PH)	Seaweed Extract (SW)
Carbon (C) %	23.36	11.34	29.97
Nitrogen (N) %	1.22	4.14	5.41
Phosphorus (P) µg g^−1^	334	15,000	12,500
Potassium (K) µg g^−1^	3970	4100	107,100
Calcium (Ca) µg g^−1^	3940	772	10,800
Magnesium (Mg) µg g^−1^	1600	269	936
Sulphur (S) µg g^−1^	4990	1730	12,500
Iron (Fe) µg g^−1^	8470	17	3310
Manganese (Mn) µg g^−1^	49	142	57
Zinc (Zn) µg g^−1^	32	12	558
Copper (Cu) µg g^−1^	22	1	523
Boron (B) µg g^−1^	22	4	944
Molybdenum (Mo) µg g^−1^	5	nd	8

**Table 2 plants-13-02188-t002:** Amino acid profile of the protein hydrolysate product.

Amino Acids	Content mM	Amino Acids	Content mM
Alanine (Ala)	58.31	Lysine (Lys)	52.84
Arginine (Arg)	4.46	Methionine (Met)	21.10
Asparagine (Asn)	18.51	Phenylalanine (Phe)	34.67
Aspartic Acid (Asp)	39.02	Proline (Pro)	6.43
Glutamic acid (Glu)	55.67	Serine (Ser)	49.42
Glutamine (Gln)	1.01	Tau proteins (Tau)	8.06
Glycine (Gly)	15.98	Threonine (Thr)	34.52
Histidine (His)	10.31	Tryptophan (Try)	1.51
Isoleucine (Ile)	30.78	Tyrosine (Tyr)	19.15
Leucine (leu)	70.46	Valine (Val)	40.74

**Table 3 plants-13-02188-t003:** Specification of Tully’s potting mix.

Characteristics	Number or Mean ± Standard Deviation
Substrate	50% bark, 25% peat, 25% pumice
Fertility and lime amendments	5 kg m^−3^ Osmocote, 1 kg m^−3^ Osmoform, 3 kg m^−3^ Dolomag, 1 kg m^−3^ gypsum, 2.5 kg m^−3^ lime
Moisture content (% volume)	37.9 ± 4.13
Bulk density (g mL^−1^)	0.39 ± 90.02
Porosity (% volume)	45.9 ± 1.02
Air space (% volume)	16.8 ± 0.33
Water holding capacity (%)	29.1 ± 0.69
pH	4.46 ± 0.25
Electrical conductivity (µg cm^−1^)	1366 ± 6
Soluble salt (%)	0.47 ± 0.22

Nutrient content of key ingredients: Osmocote: nitrate nitrogen 5.9%, ammoniacal nitrogen 7.7%, urea nitrogen 4.4% phosphorus pentoxide 9%, potassium oxide 10%, magnesium oxide 2%, iron 0.3%, manganese 0.04%, boron 0.01%, copper 0.037%, molybdenum 0.015%, zinc 0.011%. Osmoform: ammoniacal nitrogen 3.0%, urea nitrogen 7.5%, ureaformaldehyde 11.5%, phosphorus pentoxide 5%, potassium oxide 11%, magnesium oxide 2%, iron 0.5%, manganese 0.1%, boron 0.01%, copper 0.02%, molybdenum 0.001%, zinc 0.02%. Dolomag: calcium and magnesium carbonate 95–98%, calcium oxide content 28–32%, magnesium oxide content 17–20%, humidity 4% ± 2.

**Table 4 plants-13-02188-t004:** Treatments and application rates of the biostimulants.

Humic Acids (HA)					
**Description**	Extra low	Lower dose	Recommended dose from manufacturer	Higher dose	Extra high
**Application rate**	0.5 g L^−1^	1.5 g L^−1^	3 g L^−1^	6 g L^−1^	12 g L^−1^
**Abbreviation**	HA 0.5	HA 1.5	HA 3	HA 6	HA 12
**Protein Hydrolysates (PH)**					
**Description**	Extra low	Lower dose	Recommended dose from manufacturer	Higher dose	Extra high
**Application rate**	0.12 mL L^−1^	0.24 mL L^−1^	0.48 mL L^−1^	2.4 mL L^−1^	4.8 mL L^−1^
**Abbreviation**	PH 0.12	PH 0.24	PH 0.48	PH 2.4	PH 4.8
**Seaweed Extract (SW)**					
**Description**	Extra low	Lower dose	Recommended dose from manufacturer	Higher dose	Extra high
**Application rate**	0.1 g L^−1^	0.25 g L^−1^	0.5 g L^−1^	1 g L^−1^	10 g L^−1^
**Abbreviation**	SW 0.1	SW 0.25	SW 0.5	SW 1	SW 10

The recommended application rates from the manufacturer were converted to the equivalents for potted plant use, based on the surface area and size of the pot.

**Table 5 plants-13-02188-t005:** Scoring criteria of seedling growth in the in vitro experiment. The overall growth score was calculated by averaging the scores from various aspects of the plant growth criteria.

**Shoot growth and development scoring criteria**		
	*Score*	*Criteria*
Shoot length	5	Shoot length > 4 cm
	4	3 cm < Shoot length ≤ 4cm
	3	2 cm < Shoot length ≤ 3cm
	2	1.5 cm < Shoot length ≤ 2cm
	1	Shoot length ≤ 1.5 cm
	0	Shoot dead
**Leaf growth and development scoring criteria**		
Leaf number	5	Leaf number > 8
	4	6 < Leaf number ≤ 8
	3	4 < Leaf number ≤ 6
	2	2 < Leaf number ≤ 4
	1	Leaf number at 2
	0	Leaves dead and dried
Leaf size	5	Leaf diameter is mostly over 1.5 cm
	4	Leaf diameter mostly in the range of 1 cm to 1.5 cm
	3	Leaf diameter mostly in the range of 0.5 cm to 1.0 cm
	2	Leaf diameter is mostly around 0.5 cm
	1	leaf diameter is less than 0.5 cm
	0	Leaves dead and dried
**Root growth and development scoring criteria**		
Development of root hairs	5	Secondary and tertiary roots developed with massive root hairs
	4	Secondary roots developed with a medium number of root hairs
	3	Few secondary roots and sparse root hairs
	2	Little to no secondary roots and root hairs
	1	Only primary root without root hairs
	0	Root dead

## Data Availability

Data are presented in the paper. Raw data can be downloaded at: https://1drv.ms/u/s!AhAHGPOcRCGma9k0SeZdalfO5kE?e=tTLul3 (accessed on 21 May 2024).

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
