# Peer review of "Revealing the Complex Interplay of Biostimulant Applications"

_plants, 2024, doi:10.3390/plants13162188_

Round 1

Reviewer 1 Report

Comments and Suggestions for Authors

I have made comments in the body of the MS.  A few comments and critical issues:

1) Overall the greenhouse experiment is useful as it shows some interesting dose response curves, something that is not often seen.

2) Materials and methods are inadequate - how was biostimulant applied? I do not believe you can contrast the equivalency of application rate between field application and pot application - explain?  You provide no information on background nutrients in the potting mixture - this is essential if you are to interpret the possible effects of nutrients within the BS application.  Did you measure plant or soil nutrient concentrations?

3) The entire Agar Plate experiment is highly suspect. Essentially all effects were negative, many of them apparently toxic.  It is not appropriate to look for interactions between mixtures of BS's under this circumstance. It seems evident that your effective application rate in the Agar Test was toxic (as shown with excess rates in the pot trial).

4) I do not feel there is any value in the agar plate trial.  Further I do not feel any conclusions about microbial interactions and soil structural effects is warranted - results from dead and dying plants are really not relevant.

5) Much of your discussion is highly speculative and refers to a lot of proposed explanations that you did not measure.

Overall the greenhouse portion of this experiment is interesting and useful and might warrant publication if all issues can be adressed.

Author Response

Dear Reviewer:

Thanks for taking the time to review our manuscript titled “Revealing the complex interplay of biostimulant application” Submitted to Plants.

All feedback has been carefully considered, and extensive revisions have been made throughout the manuscript. Below are responses to the comments made on the paper.

Please find attached the revised manuscript with all the changes highlighted in the text. Looking forward to your further comments for the manuscript and please feel free to contact us if you require any further information or clarification.

Kind Regards,

Ye Yuan

*Please see attachment for the response to the comments

Reviewer 2 Report

Comments and Suggestions for Authors

The paper describes the results of greenhouse and laboratory research, experiments on the complex interaction of biostimulants (humic acids, protein hydrolysates, seaweed extracts) and applications using Antirrhinum majus L. as a plant pattern.  The topic is interesting and the results could provide progress in current knowledge on the use of biostimulants.  However, reviewing this manuscript and assessing its suitability for publication in the journal "RoÅ›liny" I believe that the scientific quality of the manuscript is poor and cannot be taken into account for publication in its present form.

The introduction does not contain sufficient information and does not contain appropriate references to the research.  On the basis of a review of the subject literature, the current state of knowledge on the research topic should be briefly presented.  The hypothesis put forward in the study should be clearly formulated.

The description of the materials and methods is not clear and does not contain sufficient detail to allow for replication of the study by an independent investigator.  What was the design of the experiment?  What was an experimental unit?  The number of plants in each treatment?  The number of repetitions of each treatment?  Lines 502-503: "with six repetitions of each treatment" – six repetitions of each treatment or a total of six plants (Figure 1)?  It's not clear.  If there are six plants in total, the experimental material is insufficient.  What did the follow-up treatment consist of?  Number of plants in each treatment to measure plant morphology (number of side shoots, shoot length, number of leaves, leaf area)?  Table 8 is not clear enough to read.

The results are too generic.  Lack of analysis of the survey results.  The study concludes that the results should be presented in more detail according to the ANOVA and PCA.  This self-contained (humic acids, protein hydrolysates, seaweed extracts) and combined effects (humic acids x protein hydrolysates, humic acids x seaweed extracts, protein hydrolysates x seaweed extracts, humic acids x protein hydrolysates x seaweed extracts) of the biostimulants studied should be clearly presented.  Line 72: "in specific concentration ranges" – in what concentration range?

Another important drawback is the Discussion section.  The authors should objectively and critically interpret the results on the basis of the current state of knowledge about the research.  Table 3 should be moved to the Materials and methods section.  Figure 5 should be moved to the Results section.  Lines 298-302: References must be added.

Proposals should present the main conclusions of the study and recommendations for future research, which may be based on the results of the studies.

The abstract should briefly state the purpose of the research and summarize the main results of the articles.  Line 19: "in critical concentration range" – In what concentration range?

Author Response

(The authors gave the same response as above.)

Round 2

Reviewer 1 Report

Comments and Suggestions for Authors

This is much improved in all aspects except the Materials and Methods that require more clarity and the discussion that is very poorly written and full of poorly relevant references and very poor English.

For M&M  It is critical that you explain how you arrived at concentration equivalency and how the 'field ' rate was determined. You should also explain how and why a soil application could be very different from an in-vitro application.

For the Discussion - Be much more analytical is describing how the greenhouse conditions/plant age/dilutions/irrigation/nutrients/pH etc. might have resulted in the different results.

Comments on the Quality of English Language

Author Response

Dear reviewer:

Thank you very much for your time and effort in reviewing our work, and I have modified the manuscript according to your commons.

For Material and method section: More details have been added: Line 548-549, "Throughout the experimental period-from the pots" Line 612-613, "Footnote added in Table 6". 

For the Discussion section: the discussion section has been reconstructed and partly rewritten, Line 420-421, Line 427-442.  

Some gramma mistakes and typos have been identified and corrected in the newly revised manuscript, and the manuscript currently goes through the language revision process by the co-author, but may require more time to complete the process. We will submit the final version later after confirming the content. If you have any further comments or suggestion, please feel free to contact us.

Kind Regards

Ye

Reviewer 2 Report

Comments and Suggestions for Authors

The paper describes the results of a greenhouse and laboratory research, experiments on the complex interaction of biostimulants (humic acids, protein hydrolysates, seaweed extracts) and applications using Antirrhinum majus L. as a plant pattern. The topic is interesting and the results  could provide progress in current knowledge on the use of biostimulants. I believe that the manuscript has been sufficiently improved to be considered for publication in the journal Plants. the Authors took into account all my comments.

Author Response

Dear Reviewer:

Thank you very much for your kind and encouraging review of our manuscript. I have made some adjustments to enhance the clarity of the paper.

Your time and effort in reviewing our work are highly appreciated. If you have any further comments or suggestion, please feel free to contact us.

Kind Regards

Ye